# **Open-loop GPS signal tracking at low elevation angles from a ground-based observation site**

Georg Beyerle and Florian Zus

GFZ German Research Centre for Geosciences, Potsdam, Germany

*Correspondence to:* Georg Beyerle, GFZ German Research Centre for Geosciences, Telegrafenberg, D-14473 Potsdam, Germany (gbeyerle@gfz-potsdam.de)

Abstract. A one-year data set of ground-based GPS signal observations aiming at geometric elevation angles below  $+2^{\circ}$  is analyzed. Within the "GLESER" measurement campaign about 2600 validated setting events were recorded by the "OpenGPS" open-loop tracking receiver at an observation site located at 52.3808°N, 13.0642°E between January and December 2014. The measurements confirm the feasibility of open-loop signal tracking down to geometric elevation angles of  $-1^{\circ}$  to  $-1.5^{\circ}$ 

- extending the corresponding closed-loop tracking range by up to 1°. The study is based on the premise that observations of low-elevation events by a ground-based receiver may serve as test cases for space-based radio occultation measurements, even if the latter proceed at a significantly faster temporal scale. The results support the conclusion that the open-loop Doppler model has negligible influence on the derived carrier frequency profile for strong signal-to-noise density ratios above about 30 dB Hz. At lower signal levels, however, the "OpenGPS" receiver's dual-channel design, which tracks the same signal using
- two Doppler models with a 10 Hz offset, uncovers a notable bias. The repeat patterns of the GPS orbit traces in terms of azimuth angle reveal characteristic signatures in both, signal amplitude and Doppler frequency with respect to the topography close to the observation site. On the other hand, vertical refractivity gradients extracted from ECMWF meteorological fields correlate moderately well with observed signal amplitude fluctuations at negative geometric elevation angles emphasizing the information content of low-elevation GPS signals with respect to the atmospheric state within the planetary boundary layer.

# 15 1 Introduction

For more than a decade space-based Global Navigation Satellite System (GNSS) radio occultation (GNSS-RO) has established itself as a valuable measurement technique for atmospheric remote sensing. Vertical profiles of ray bending angle, refractivity, dry pressure and temperature are used by several meteorological centres for assimilation into numerical weather prediction models (see, e.g., Cucurull et al., 2007; Anthes et al., 2008; Healy and Thepaut, 2006; Liu and Xue, 2014; Poli et al., 2010;

Rennie, 2010, and references therein). Moreover, climate studies increasingly take advantage of validated GNSS-RO data sets (see, e.g., Foelsche et al., 2011; Ringer and Healy, 2008; Steiner et al., 2011; Gleisner and Healy, 2013; Schmidt et al., 2010; Poli et al., 2010, and references therein).

A number of past and current spacecrafts carry GNSS-RO payloads, e.g. the satellites GPS/Met (Kursinski et al., 1997), CHAMP (Wickert et al., 2001), GRACE (Beyerle et al., 2005; Wickert et al., 2005), COSMIC (Anthes et al., 2008), Metop

(Luntama et al., 2008; von Engeln et al., 2011; Bonnedal et al., 2010; Zus et al., 2011), TerraSAR-X (Beyerle et al., 2011), TanDEM-X (Zus et al., 2014). Already the proof-of-concept mission GPS/Met revealed the difficulties of retrieving dual frequency carrier phase data when the ray tangent point enters the lower troposphere (Rocken et al., 1997). More specifically, Rocken et al. (1997) noticed a significant negative refractivity bias in the lower troposphere at tropical latitudes. At low altitudes the GNSS signals experience multipath beam propagation (see, e.g., Gorbunov, 2002; Hocke et al., 1999). The resulting optical

the GNSS signals experience multipath beam propagation (see, e.g., Gorbunov, 2002; Hocke et al., 1999). The resulting optical path length differences lead to constructive and destructive interferences and the corresponding signal amplitude fluctuations increase the probability of an early loss of tracking lock.

To address these issues new signal tracking methods were developed and implemented. Whereas the "fly-wheeling" tracking method of JPL's "Blackjack" GPS receivers mounted on CHAMP and GRACE (Hajj et al., 2004), showed some progress,

significant improvements with respect to probing of the planetary boundary layer, in particular at low latitudes, were obtained with the introduction and implementation of open-loop (O/L) tracking (or raw-sampling) techniques (see, e.g., Sokolovskiy, 2001; Sokolovskiy et al., 2006; Ao et al., 2009; Bonnedal et al., 2010).

The open-loop signal tracking mode successfully resolves the problem of premature loss of signal in the lower troposphere at low latitudes (Sokolovskiy, 2001; Sokolovskiy et al., 2006). In contrast to closed-loop (C/L) tracking, a receiver operating in

- open-loop mode partially or completely disregards the tracking loop feedback values from the carrier and code discriminators, but instead steers the corresponding numerically-controlled oscillators (NCOs) using a-priori parameters. In the following these O/L parameters, which are usually derived from an atmospheric climatology, are referred to as "O/L model". The time duration, which the receiver operates in open-loop tracking mode, may be controlled by predetermined threshold values in terms of tangent point altitude, elevation angle and/or signal-to-noise ratio (SNR). While these general considerations apply to
- all open-loop / raw-sampling implementations, the specific realizations vary in detail (see, e.g., U.S. patents US6731906 B2 and US6720916 B2).

A key requirement for the adoption of O/L signal tracking in operational GNSS-RO missions is the insensitivity of derived carrier phase paths and code pseudoranges on the particular choice of O/L model. According to Sokolovskiy (2001) the requirement is met, provided the true atmospheric Doppler profile deviates by not more than half of the sampling frequency from

the O/L Doppler model. With a typical sampling rate of  $f_s = 50$  Hz this requirement translates into a maximum frequency deviation of 25 Hz.

Recently, Beyerle et al. (2011) claimed on the basis of GNSS-RO observations recorded by the "IGOR" receiver aboard the TerraSAR-X spacecraft, that the O/L Doppler model may influence the derived refractivity values for low signal amplitudes below about 25 V/V and potentially contribute to the negative refractivity bias. In order to substantiate this hypothesis

- they proposed to track each GNSS-RO signal with two O/L models, separated in frequency space by a predefined offset. We note, however, that other causes certainly contribute to the observed refractivity bias as well (see, e.g., Gorbunov et al., 2015; Sokolovskiy et al., 2010, and references therein). Since "IGOR" firmware modification aboard TerraSAR are unfeasible, two indirect methods were investigated. First, a measurement campaign was conducted in late 2012 with the GNSS-RO receivers aboard the TerraSAR-X and TanDEM-X satellites recording signals from the same setting GPS satellites, but using different
- O/L models (F. Zus et al. (2014), "GPS radio occultation with TerraSAR-X and TanDEM-X: sensitivity of lower troposphere

sounding to the Open-Loop Doppler model", Atmos. Meas. Tech. Discuss., 7, 12719–12733, doi:10.5194/amtd-7-12719-2014, unpublished). Second, within the framework of the "GLESER" (GPS low-elevation setting event recorder) measurement campaign a ground-based experiment was devised and established at an observation site on the "Albert Einstein" science campus in Potsdam, Germany ( $52.3808^{\circ}N$ ,  $13.0642^{\circ}E$ ) in December 2010. The "GLESER" campaign targets signals from GPS satellites at low elevations as they set beyond the horizon at elevation angles below  $+2^{\circ}$ . The measurement hardware, a single-frequency "OpenGPS" receiver, is an in-house development based on C. Kelley's "OpenSourceGPS" concept (Kelley, 2002).

Whilst ground-based observations of low-elevation setting events do not allow to derive bending angle profiles for ray tangent points above the receiver altitude (Zuffada et al., 1999; Haase et al., 2014; Healy, 2002; Sokolovskiy et al., 2001), these measurements nevertheless are useful to investigate receiver tracking behaviour under multipath conditions with strongly

- fluctuating SNRs. In addition, the signal excess phase paths have been shown to be sensitive to the local refractivity field (see, e.g., Lowry et al., 2002; Zus et al., 2015). During a typical low-elevation setting event the geometric elevation angle, i.e. the ray's elevation angle at the receiver antenna, disregarding atmospheric refraction effects, decreases from  $+2^{\circ}$  at the measurement start to about  $-1^{\circ}$  to  $-1.5^{\circ}$  at the end of the observation. Setting events last for about 10 to 15 minutes on average; hence their durations are about an order of magnitude longer than typical space-based radio occultation measurements
- (see, e.g. Kursinski et al., 1997). Even if a ground-based observation does not lend itself to the derivation of bending angle profiles (Zuffada et al., 1999; Haase et al., 2014; Healy, 2002; Sokolovskiy et al., 2001), from the signal tracking perspective we may still regard "GLESER" recordings as radio occultation events in slow motion. The present study is restricted to the observation and analysis of setting events; an extension towards rising events, however, is feasible from a technical standpoint and may be considered in the future.
- The paper is sectioned as follows. First, GFZ's "OpenGPS" instrument, which has participated in several ground-based and airborne measurement campaigns during the last decade, is described. Closed-loop and open-loop tracking methods are briefly reviewed and the receiver's capabilities are illustrated with some example profiles. The section ends with a cursory review of the "OpenGPS" hardware and software. Second, the measurements conducted during the "GLESER" campaign are introduced and the data processing algorithms and analysis methods are discussed. In the final and main section of this paper the measurement results are discussed and put into perspective.

the measurement results are discussed an

### 2 Instrument description

The "GLESER" campaign utilizes the "OpenGPS" instrument, a single-frequency 12 channel GPS receiver. Several copies of this device, which is based on C. Kelley's "OpenSourceGPS" concept (Kelley, 2002), were built at GFZ and used in various ground-based and airborne GPS measurement campaigns (see, e.g., Helm, 2008, and Helm et al., 2004, "Detection of coherent

reflections with GPS bipath interferometry", unpublished, preprint available at http://arxiv.org/abs/physics/0407091). In order to provide a self-consistent description of the "OpenGPS" instrument and its O/L signal tracking implementation, we begin with a brief review of C/L and O/L tracking techniques.

#### 2.1 Closed-loop and Open-loop Signal Tracking

It well known that inhomogeneities in the tropospheric water vapour field, in particular at low latitudes, can produce multipath propagation of GNSS signals at low elevation angles (see, e.g., Gorbunov et al., 2004; Beyerle et al., 2003). Space-based RO observations show that under these conditions SNR values exhibit strong fluctuations, which early GNSS-RO receivers were 5 unable to track properly (Rocken et al., 1997). To address premature signal loss GNSS-RO receiver tracking algorithms based on closed feed-back loops were replaced by "open-loop" techniques (see, e.g., Sokolovskiy, 2001; Sokolovskiy et al., 2006; Ao et al., 2009; Bonnedal et al., 2010). If a receiver operates in open-loop mode the feed-back loop is opened and the NCO producing the replica signal is steered (in part or fully) from model parameters. This model takes into account the expected signal dynamics from both, the transmitter and receiver orbits, clocks biases and drifts as well as the signal propagation characteristics in the lower troposphere. The latter are typically obtained from an atmospheric climatology (Sokolovskiy, 10

2001; Bonnedal et al., 2010).

The schematic in fig. 1 illustrates the two tracking concepts for carrier phase tracking; corresponding considerations apply to code tracking as well. In standard C/L tracking (fig. 1, top panel) the down-converted input signal ("input") is correlated with two internal replica signals ("sin" and "cos") generated by the NCO (see, e.g., Misra and Enge, 2006). The result is low-pass

- filtered (represented by the box labeled "average"), the carrier phase discriminator determines phase deviations between the 15 observed and modeled replica and provides appropriate adjustments to the loop filter. The "sin" and "cos" replica signals, the latter being phase shifted by a quarter cycle, i.e.  $90^{\circ}$  with respect to the former, allow to distinguish phase advances from phase delays between observed and replica signal. The phase discriminator output is digitally filtered ("loop filter") to prevent unstable loop behaviour (see, e.g., Lindsey and Chie, 1981; Thomas, 1989). The receiver output samples ("carrier
- phase output") combine the NCO model phases and the phase residuals from the discriminator to yield the observed carrier phase.

If signal amplitudes drop below certain threshold levels, the corresponding phase residuals start to be dominated by noise and proper alignment between replica and observed signal can no longer be maintained. Fig. 2 shows exemplarily the signalto-noise density ratio  $C/N_0$  (Badke, 2009; Kaplan, 1996; Parkinson and Spilker, 1996) as a function of geometric elevation angle for a setting event recorded in the morning of 1 January 2015. At elevation angles below about  $+1.5^{\circ}$  the density ratios

- $C/N_0$  start to fluctuate. At about +1.19° and again at +0.56° transient signal gaps with  $C/N_0 \lesssim 30$  dB Hz occur which last for less than about 0.5 s. During these time intervals the phase discriminator output is dominated by noise causing enhanced NCO Doppler frequency fluctuations as is illustrated in fig. 3 (blue line). At about  $-0.26^{\circ}$  elevation low signal level conditions persist for a longer time period and the carrier NCO frequency deviates by hundreds of Hz (blue line in fig. 3). Correspondingly, C/N<sub>0</sub> drops by about 20 dB down to the noise level (fig. 2) and never recovers during the last part of this setting event.

Open-loop tracking is immune to transient SNR gaps, even if these breaks stretch across extended time periods. The bottom panel of fig. 1 schematically illustrates the concept. In O/L tracking mode the feedback loop is removed and the NCO is solely controlled by model values. The correlation output values, produced by the "cos" and "sin" branches, are denoted by "in-phase" and "quad-phase" correlation samples, respectively (Sokolovskiy, 2001; Sokolovskiy et al., 2006; Ao et al., 2009;

**Figure 1.** Schematic representation of closed-loop (top panel) and open-loop (bottom) carrier signal tracking. In closed-loop mode the "input" signal is correlated with two replica signals, "sin" and "cos", the latter is phase-shifted by 90° with respect to the former. The correlation sums are low-pass filtered and the output is examined for phase deviations between input and replica signal. The discriminator adjustments close the feedback loop. Open-loop tracking (bottom panel) dispenses with the phase discriminator and the numerically controlled oscillator ("NCO") is solely steered from model values ("Doppler model"). The observed carrier phase finally is assembled from the NCO phases and the in-phase and quad-phase correlation samples.

Bonnedal et al., 2010). In low-SNR conditions the in-phase and quad-phase samples are dominated by noise. However, since no feedback is present in O/L tracking, these noisy samples cannot produce erroneous control input to the NCO and transient signal gaps do not cause loss of tracking lock as illustrated in figs. 2 and 3. Red and green lines show C/N<sub>0</sub> (fig. 2) and the NCO carrier frequency (fig. 3) recorded by the receiver's two open-loop channels, respectively. In the following the two channels

5