# Peer review of "Open-loop GPS signal tracking at low elevation angles from a ground-based observation site"

_Atmospheric Measurement Techniques, 2016_

## Referee Comment (RC1) · Anonymous Referee #1 · 22 Jun 2016

**Review of paper "Open-loop GPS signal tracking at low elevation angles from a ground-based observation site" by Georg Beyerle and Florian Zus**

**General Remarks**

The paper describes ground based observations of GPS signals tracked in the open-loop mode by the multi-channel OpenGPS hardware developed and built at GFZ. The experimental allows for tracking signals at high sampling rates and saving all the raw data for the further analysis. The tracking was performed by two channels using NCO models separated by 10 Hz.

The paper present an extremely interesting experimental study, and it definitely deserves publishing.

On the negative side, the paper is overloaded with technical details. For example, for a reader it may be difficult and hardly necessary to understand DUMP and TIC events. The authors should more concentrate on the issues important for radio occultations. On the other hand, the paper discusses multipath propagation, but it suffers from the lack of discussion of difference between the conditions of multipath propagation for space-born and ground-based receiver observations. Therefore, the paper should be stripped off the technical details and complemented with a physical discussion of multipath propagation in ground-based observations.

**Specific Remarks**

*1. Introduction*
While the authors are discussing ground-based experiment, it is expedient to discuss not only radio occultations, but also ground-based observation used for integrated water vapor retrieval.

36–37. *The resulting optical path length differences lead to constructive and destructive interferences*
Terms "constructive interference" and "destructive interference" are understood as in-phase or counter-phase interference resulting in amplification of attenuation of resulting amplitude. Such a detailed description is unnecessary here. It is enough to say that interference results in signal scintillations.

92–93. *...from the signal tracking perspective we may still regard "GLESER" recordings as radio occultation events in slow motion.*
This statement needs more elaboration. Ground-based observations have a different geometry and, therefore, different conditions for multipath propagation. These factors have to be analyzed.

163–165. *This 10 Hz shift can clearly be identified in fig. 3 (insert); here, O/L tracking mode starts at an elevation angle of −0.08° and reaches the nominal 10 Hz shift after a short settling phase at −0.13° elevation.*
However, here we observe some asymmetric between red and green curve. How can this be accounted for?

170–180. For a reader, it may be difficult to understand what DUMP and TIC events are. Is it necessary to go into these technical details?

*Figure 9. Success rate of internal navigation bit retrieval…*
I would not term the value of E a success rate. This value is zero for the ideal demodulation, and it is unity for randomly chosen navigation bits. Therefore, it should rather be termed a failure rate.

393. *with respect azimuth angle*
with respect to azimuth angle

Figure 13. Add the explanation of dark and light blue curves in the caption.

515–525, Figure 16.
All this part needs a significant improvement. The authors state that multipath presupposes vertical refractivity gradient, which is simply wrong. Take a simple exponential atmospheric model. This model acts as a defocusing lens, and, regardless of the vertical refractivity gradient, never creates multipath (effects like planetary flash are put aside). Multipath propagation is caused rather by non-monotonic profile of refraction angle, i.e. it should be linked to the second derivative of the refractivity. However, even this would not suffice, because this effect requires some observation distance. The authors state that they average the vertical gradient over the height interval from 0 to 2 km, but it remains unclear, what horizontal locations were chosen. If they took just the observation region, then this does not make sense. Figure 16 is not convincing either. In my view, none of the plots indicates any correlation between the average refractivity gradient and C/N0 fluctuations. I cannot see anything special for PRN 22, 7, and 13. Moreover, this is not surprising in the light of my previous remark.
The authors should present a physical analysis of the multipath propagation condition, instead of the speculative model. Consequently, Figure 16 should be replaced by the correlation with a more characteristic quantity derived from ECMWF fields. The best way is just to perform forward simulation. To my knowledge, at least one of the authors (Georg Beyerle) has enough experience for that.

---

## Referee Comment (RC2) · Anonymous Referee #2 · 24 Jun 2016

In this paper, the authors discuss the tracking performance of ground GPS observations at low elevation angles using their implementation of the OpenGPS receiver and suggest that the ground-based measurements provide a useful test bed for spaceborne radio occultation. I appreciate the concept and think that this can be a worthwhile approach. However, a better experimental design and perhaps better site selection are needed to produce more impactful results. The main conclusions of this paper are not really new, at least qualitatively. In addition, one issue I have is with the implementation of its O/L tracking (see comment 3 below) and how this would affect the results quantitatively and how this would translate to the spaceborne case. Overall, I think the paper can be suitable for publication, provided that the comments are adequately addressed

and that the paper is viewed as a "pathfinder" for future experiments.

(1) The paper should be made more succinct. In particular, the detailed description of the OpenGPS receiver does not offer much insight into understanding the data. The readers have to wade through too much background materials before getting to the results. I suggest significant shortening of Section 2.2 and Section 2.3. Another option is to move the materials to an Appendix.

(2) L15: it states that "vertical refractivity gradients... correlate moderately well with observed signal amplitude fluctuations..." However, Figure 16 shows that only 3 out of 9 cases "yield significant correlations". I suggest changing the wording of "correlate moderately well" to better represent the results.

(3) L245-255: the O/L tracking models are extrapolated from the C/L tracking data at higher elevation angles. Thus one can argue that this is more similar to the "flywheel-ing" algorithm implemented on CHAMP and SAC-C and might not work well under some conditions (which the authors recognized, see L267-270). Please explain/justify why it is done this way. Is it possible to use an a priori O/L model as was done in COSMIC and Metop/GRAS?

(4) Eqs (6)-(8): is there a reference for calculating C/N0 this way? I am confused with Eq. (8) since this apparently yields a noise floor of 17 dB Hz that depends only on the integration time. Shouldn't this depend on the antenna gain, cable loss, etc.?

(5) Table 1: PRN28 yields anomalously low ($\sim 35\%$) O/L enhancement. Any idea why?

(6) Fig. 15: I suggest more distinct colors for the light blue and dark blue lines. I have a hard time distinguishing them. It is also hard to tell the actual values of frequency offsets from these plots. I think it would be useful to have a summary table of mean and standard deviation of $\Delta f_{obs}$ as a function of C/N0 that average over all 9 PRNs.

(7) L522: Is the refractivity based on ECMWF at a grid point closest to the receiver?

How about temporal differences? How does the refractivity vary along the ray paths? Besides vertical refractivity, horizontal inhomogeneity and small-scale irregularities (turbulence) can also lead to strong signal fluctuations. These effects could perhaps explain the lack of correlations for many of the PRNs.

(8) Do you expect local environmental effects (e.g., local multipath) and ionospheric conditions to have significant impact on the measurements?

---

## Author Comment (AC1) · 9 Sep 2016

**Response to reviewer 1**

We thank the reviewer for his/her time, the thoughtful suggestions and helpful comments. Guided by the two reviewers' remarks the paper has been corrected and revised. In the following we provide point-by-point responses to the reviewer's suggestions. His/her remarks are set in italics, our answers are added in normal font. The revised text is indented.

*General Remarks*
*[...]*
*On the negative side, the paper is overloaded with technical details. For example, for a reader it may be difficult and hardly necessary to understand DUMP and TIC events. The authors should more concentrate on the issues important for radio occultations. On the other hand, the paper discusses multipath propagation, but it suffers from the lack of discussion of difference between the conditions of multipath propagation for spaceborne and ground-based receiver observations. Therefore, the paper should be stripped off the technical details and complemented with a physical discussion of multipath propagation in ground-based observations.*

We concur that the paper comprises a certain amount of technical detail. For reasons discussed below, we prefer to keep sections 2.2 and 2.3 ("OpenGPS receiver hardware" / "OpenGPS receiver software"), but — following a suggestion by reviewer 2 — these two sections are now relegated to the appendix.

One of the objectives of this study is to heighten the awareness, that scientific GNSS data processing already starts at the instrument level. Thus, technical details of the receiver-internal processing algorithms may affect the final results as much as the scientific algorithms applied during post-processing.

The "OpenGPS" instrument is one of the few open source / open architecture GPS receivers currently available to the scientific community. (The "OpenGPS" receiver software and the "GLESER" raw data can be retrieved via the Digital Object Identifier (DOI) doi:10.5880/GFZ.2016.1.1.002.) Since there exists no refereed literature on the "OpenGPS" instrument, which has been used in several measurement campaigns during the last decade, we consider it worthwhile to include a certain amount of technical detail in the present paper to help those, who intend to work more closely with the "OpenGPS" source code.

Apart from the motivation to introduce "OpenGPS", another key objective is to draw the attention to the information content of GNSS signals at low elevation angles. Multipath is certainly an important aspect of this type of ground-based GNSS observation; its detailed study, however, is beyond the

scope of this paper and will be addressed in future work. Following the reviewer's suggestion (see below) we performed provisional Multiple Phase Screen (MPS) simulations to support the interpretation of the observed $C/N_0$ profiles, in particular the observed $C/N_0$ fluctuations. The revised paper includes an additional section and figures describing the MPS results (see response to specific comment below).

Differences of multipath signal propagation in spaceborne and ground-based observations is addressed in the revised section 1 ("Introduction") which includes the following additional paragraph:

> Clearly, the occurrence of multiple signal paths ("multipath") connecting transmitter and receiver instrument distinguishes ground-based from space-based GPS measurements. Multipath, caused by signal reflections in the direct vicinity of the receiving antenna, cannot be avoided in most ground-based observations (see, e.g., Parkinson and Spilker, 1996; Hofmann-Wellenhof, 2013); on a spaceborne platform, however, these local reflections may be eliminated to a large extent by careful spacecraft design and suitable antenna placement (cf., however, Gaylor et al., 2005). Within the framework of space-based radio occultation the term "multipath" assumes an alternative connotation and generally refers to tropospheric signal propagation close to the ray tangent point, thousands of kilometers away from the receiver. Local multipath, i.e. reflections in the vicinity surrounding the antenna, can be described using geometric optics (see, e.g., Elósegui et al., 1995; Anderson, 2000; Larson et al., 2008); tropospheric multipath is a diffraction phenomenon and requires wave optical methods to analyze quantitatively (see, e.g., Gorbunov, 2002; Sokolovskiy, 2001b). In the following, however, we will argue that for elevation angles below about $+2°$ wave optical effects may contribute in ground-based observations as well.

*Specific Remarks*

*1. Introduction*
*While the authors are discussing ground-based experiment, it is expedient to discuss not only radio occultations, but also ground-based observation used for integrated water vapor retrieval.*

The revised paper now starts with following paragraph in section 1 "Introduction":

> For more than a decade the existing Global Navigation Satellite System (GNSS) infrastructure is exploited in meteorological applications and climate studies. Ground-based GNSS observations reveal valuable information on the tropospheric water vapour content integrated

along the signal path (see, e.g., Bevis et al., 1994; Braun et al., 2001; Businger et al., 1996; Dick et al., 2001; Emardson et al., 1998; Rocken et al., 1997; Tregoning et al., 1998; Vedel and Huang, 2004; Ware et al., 1997; Hagemann et al., 2003, and references therein). Furthermore, space-based platforms equipped with GNSS receivers allow for the derivation of vertical profiles of atmospheric refractivity, dry pressure and temperature (see, e.g., Yunck et al., 2000; Kursinski et al., 2000; Melbourne, 2004, and references therein). GNSS data products derived from these ground-based as well as space-based observations are being used by meteorological centres for assimilation into numerical weather prediction models (see, e.g., Cucurull et al., 2007; Anthes et al., 2008; Healy and Thepaut, 2006; Liu and Xue, 2014; Poli et al., 2010; Rennie, 2010, and references therein). In addition, climate studies increasingly take advantage of validated GNSS and GNSS-RO data sets (see, e.g., Foelsche et al., 2011; Ringer and Healy, 2008; Steiner et al., 2011; Gleisner and Healy, 2013; Schmidt et al., 2010; Poli et al., 2010, and references therein).

*36–37.*
*The resulting optical path length differences lead to constructive and destructive interferences Terms "constructive interference" and "destructive interference" are understood as in-phase or counter-phase interference resulting in amplification of attenuation of resulting amplitude. Such a detailed description is unnecessary here. It is enough to say that interference results in signal scintillations.*

We concur; the revised sentence now reads:

> The resulting optical path length differences lead to signal scintillations and these amplitude fluctuations increase the probability of an early loss of tracking lock.

*92–93.*
*"[...] from the signal tracking perspective we may still regard "GLESER" recordings as radio occultation events in slow motion."*
*This statement needs more elaboration. Ground-based observations have a different geometry and, therefore, different conditions for multipath propagation. These factors have to be analyzed.*

We agree that "radio occultation in slow motion" is a somewhat loose terminology. Therefore the sentence

> Even if a ground-based observation does not lend itself to the derivation of bending angle profiles [...], from the signal tracking perspective

we may still regard "GLESER" recordings as radio occultation events in slow motion.

is replaced by

Even if ground-based observations do not lend themselves to the derivation of bending angle profiles [...], we regard "GLESER" observations as a useful tool to investigate open-loop signal tracking in environments with strongly fluctuating SNR.

*163–165.*
*"This 10 Hz shift can clearly be identified in fig. 3 (insert); here, O/L tracking mode starts at an elevation angle of $-0.08°$ and reaches the nominal 10 Hz shift after a short settling phase at $-0.13°$ elevation."*
*However, here we observe some asymmetric between red and green curve. How can this be accounted for?*

The answer to this excellent question is twofold:

First, the strong transient asymmetry between the red and green line with respect to the Doppler model, between $-0.08°$ and $-0.13°$ elevation angle, is due to the specific design of the GP2021 hardware correlator. (We considered these aspects too technical for the general reader and therefore omitted a more detailed explanation.) The GP2021 hardware correlator allows for frequency adjustments of the NCOs in units of $\Delta f_{\rm NCO} \equiv 42.6$ mHz; it does, however, not support phase adjustments. I.e., during the first few seconds of O/L tracking the two clone channels are gradually shifted towards the *phase* of the O/L model by changing the corresponding NCO *frequency* by $\pm\Delta f_{\rm NCO}$ until the NCO is phase aligned with the O/L model. For the setting event shown in fig. 3 this alignment process starts at about $-0.08°$ elevation angle and is completed at about $-0.13°$. The fact that in this particular case the initialization process for the $+5$ Hz channel (red) is completed much earlier than for the $-5$ Hz channel (green) is accidental; the red channel happened to be already in good alignment at the start of the initialization.

Second, disregarding the 10 Hz offset between the two O/L channels, residual fluctuations on the order of some 100 mHz are apparent even upon completion of the initialization phase (elevation angle below $-0.13°$). Again, these deviations arise from the specific design chosen for the "OpenGPS" O/L implementation. The GP2021 hardware correlator supports multichannel carrier frequency (and code) adjustments, i.e. corrections to more that one channel can be applied concurrently using the "MULTI_CHANNEL_SELECT" register. However, in our tests we found, that using this functionality had

adverse side effects and therefore a separate feedback loop was implemented and is used to keep the two clone channels separated by 10 Hz in Doppler space. The feedback loop's finite bandwidth cause the non-zero residual fluctuations apparent in fig. 3 (insert).

The revised paper now includes the following explanation (figure number 3 refers to the paper, not this document):

> Close inspection of the O/L carrier NCO frequencies in fig. 3 (red and green lines), the insert shows a zoomed-in view, reveals residual deviation of the O/L channel from the target Doppler model in addition to the nominal values of $\pm 5$ Hz. First, at the start of O/L tracking mode (at about $-0.08°$ elevation angle) the two clone channels are gradually moved into phase alignment to the O/L model by adjusting the corresponding NCO frequencies (the GP2021 hardware correlator does not support carrier phase adjustments). The duration of this initialization process is shorter if the (arbitrary) initial phase happens to be closer to the model phase targeted. Furthermore, even after completion of the alignment process, small fluctuations on the order of a few 100 mHz are still apparent. These fluctuations are caused by the finite bandwidth of a dedicated feedback loop, which keeps the two O/L channels separated by 10 Hz in Doppler space. (Accessing the GP2021's "MULTI_CHANNEL_SELECT" registers led to adverse side effects in our tests.)

*170–180.*
*For a reader, it may be difficult to understand what DUMP and TIC events are. Is it necessary to go into these technical details?*

The discussion of DUMP and TIC events is admittedly somewhat technical. Following a suggestion by reviewer 2 sections 2.2 and 2.3 ("OpenGPS receiver hardware" / "OpenGPS receiver software") are now relegated to the appendix.

*Figure 9.*
*"Success rate of internal navigation bit retrieval..."*
*I would not term the value of E a success rate. This value is zero for the ideal demodulation, and it is unity for randomly chosen navigation bits. Therefore, it should rather be termed a failure rate.*

The point is well taken; text and figure caption are changed accordingly.

*393.*
*"with respect azimuth angle" with respect to azimuth angle*
*Figure 13.*

*Add the explanation of dark and light blue curves in the caption.*

The typo is fixed and caption to fig. 13 (this figure number refers to the paper, not this document) now reads:

> Same as fig. 10, however, showing the residual frequency in O/L channel A as a function of $\Delta C/N_0$, the difference of the two O/L density ratios (gray points). The theoretically expected result (equation 15) is shown as dark blue lines. A significant amount of frequency aliasing is apparent from clusters following the light blue lines which mark the expected result (equation 15), but are shifted by $\pm 50$ Hz. The horizontal dashed lines indicate the $-5$ Hz residual frequency.

*515–525, Figure 16.*
*All this part needs a significant improvement. The authors state that multipath presupposes vertical refractivity gradient, which is simply wrong. Take a simple exponential atmospheric model. This model acts as a defocusing lens, and, regardless of the vertical refractivity gradient, never creates multipath (effects like planetary flash are put aside). Multipath propagation is caused rather by non-monotonic profile of refraction angle, i.e. it should be linked to the second derivative of the refractivity.*

*However, even this would not suffice, because this effect requires some observation distance. The authors state that they average the vertical gradient over the height interval from 0 to 2 km, but it remains unclear, what horizontal locations were chosen. If they took just the observation region, then this does not make sense. Figure 16 is not convincing either. In my view, none of the plots indicates any correlation between the average refractivity gradient and C/N0 fluctuations. I cannot see anything special for PRN 22, 7, and 13. Moreover, this is not surprising in the light of my previous remark.*

*The authors should present a physical analysis of the multipath propagation condition, instead of the speculative model. Consequently, Figure 16 should be replaced by the correlation with a more characteristic quantity derived from ECMWF fields. The best way is just to perform forward simulation. To my knowledge, at least one of the authors (Georg Beyerle) has enough experience for that.*

The reviewer is correct, that a non-zero vertical refractivity gradient is a necessary, but certainly not a sufficient condition for the occurrence of multipath. The revised paper includes a new section describing Multiple Phase Screen (MPS) simulations suggested by the reviewer:

**Simulations**

In order to support the interpretation of the observed C/N$_0$ fluctuations we performed a series of Multiple Phase Screen simulations (Knepp, 1983; Martin and Flatté, 1988; Grimault, 1998). The propagation of a plane wave through the lower troposphere is modeled by a series of 500 non-equidistant phase screens ranging from the receiver location to a distance of 500 km. On each phase screen the wave suffers a phase delay determined by the interscreen distance $\Delta z$ and the refractivity height profile (Sokolovskiy, 2001a)

$$ N(h) = 400 \exp\left(\frac{-h}{h_s}\right) \left(1 - 0.05 \frac{2}{\pi} \operatorname{atan}\left(\frac{h - h_{tp}}{h_{zn}}\right)\right) \qquad (1) $$

with scale height $h_s = 8$ km, planetary boundary layer (PBL) top height $h_{tp}$ and PBL top transition zone $h_{zn} = 50$ m. The interaction of the wave with the ground surface is modelled by applying a raised-cosine filter with a 6 dB steepness of 25 m (i.e., within 25 m the filter weight decreases by 6 dB) at zero altitude. The phase screens extend from $-20$ km to $+20$ km with a 5 km wide raised-cosine filter applied at the upper and lower boundary to suppress spurious diffraction effects; the receiver altitude is taken to be 50 m. The variation of elevation angle between $-2°$ and $+2°$ is modelled by tilting the ground surface and its overlying atmosphere correspondly.

Results from four simulation runs are shown in fig. 1; it displays the normalized signal amplitude as a function of elevation angle. Signal absorption at the ground surface produces characteristic diffraction patterns for elevation angels below $0°$ (red and blue lines). Without ground absorption the diffraction patterns almost disappear and the profiles resemble step functions expected from geometric optics (green and black). The simulations did not produce C/N$_0$ fluctuations for horizontally oriented PBL tops (parameterized by $h_{tp}$ in eqn. 1). However, if the top layer tilts towards the receiver, substantial signal deviations at elevation angles above $0°$ are observed. Fig. 1 illustrates this phenomenon for a PBL top layer ascending from 1 km at 30 km distance to 2 km at about 60 km (green and blue lines); below 30 km and above 60 km $h_{tp}$ remains fixed at 1 km and 2 km, respectively.

The MPS simulation results plotted in fig. 1 indicate ground effects below about $0°$ elevation angle (blue and red) and PBL-induced C/N$_0$ variations above about $0°$ (blue and green). The results suggest that these C/N$_0$ fluctuations are independent from each other and tend to be separated in elevation angle space. Finally, we note that the addition of irregularities on spatial scales characteristic for turbulence to the refractivity profiles did not produce significant C/N$_0$ changes.

On the basis of these simulation results the analysis involving the correla-

[Figure]

Figure 1: Normalized signal amplitudes as a function of elevation angle derived from several Multiple Phase Screen simulations. Refractivities on the individual phase screens are calculated from an exponential profile furnished with a planetary boundary layer in the lower troposphere. Signal absorption at the surface is taken into account (red and blue lines), green and black lines show the result without ground absorption. Two boundary layers are modelled: a horizontal boundary layer top at 2 km altitude (red and black) and a layer top increasing from 1 km to 2 km between 30 and about 60 km distance from the receiver (blue and green). For legibility the red, green and blue lines are shifted by an additional +1 dB, +2 dB and +3 dB offset, respectively. For details see text.

tion of $C/N_0$ fluctuations with tropospheric refractivity gradients extracted from ECMWF data was repeated. The revised analysis was modified in the following way:

- For the original analysis the ECMWF grid point closest to the observation site at (52°N, 13°E) was used. This grid point is about 42 km south of the receiver location. The "GLESER" observation window, however, points mainly to the west and the MPS simulations suggest that the relevant tropospheric air volumes are located about 30 to 50 km west of the observation site. The revised analysis, therefore, utilizes the ECMWF grid points (52°N, 12°E) for azimuth angles $< -90°$ and (53°N, 12°E) for azimuth angles $\geq -90°$.

- The MPS simulations also suggest that at the lowest elevation angles diffraction effects caused by the ground surface dominate the observed $C/N_0$ fluctuations. The revised analysis, therefore, restricts the ranges

of elevation angle for the estimation of $\sigma\,(\mathrm{C/N_0})$, the standard deviation of $\mathrm{C/N_0}$, to $+1°, \ldots, +2°$.

- For the revised analysis the ECMWF data set has been extended and now covers the period from 1 March 2014 to 31 August 2014.

The discussion of the revised analysis reads as follows:

> The MPS simulations (fig. 1) suggest that at negative elevation angles diffraction effects caused by the ground surface dominate the observed $\mathrm{C/N_0}$ fluctuations. At higher elevations atmospheric multipath seems to be more relevant. This hypothesis is tested for the six month time period from March to August 2014 by correlating the standard deviation of $\mathrm{C/N_0}$ between elevation angles of $+1°$ and $+2°$ with the mean refractivity gradient $\langle dN/dz \rangle$. The calculation of $\langle dN/dz \rangle$ is restricted to the altitude range from 1 to 3 km. Fig. 2 shows the results for nine PRNs.
>
> The vertical refractivity profiles $N(z)$ are extracted from European Centre for Medium-Range Weather Forecasts (ECMWF) meteorological fields. Their horizontal resolution is $1° \times 1°$ (about 110 km in meridional and 69 km in zonal direction at the receiver location) with 137 height levels ranging from 0 to about 80 km; the averaging interval of 1 to 3 km corresponds to about 13 vertical height levels. For signal azimuth angles less than $-90°$ (west to south-west) the refractivity profile is extracted from ECMWF grid point (52°N, 12°E), about 84.4 km south-west of the observation site ($-119.8°$ true bearing). For azimuth angles greater than $-90°$ (west to north-west) the ECMWF grid point (53°N, 12°E) is selected, which is located about 99.8 km in the north-western direction ($-45.8°$ true bearing). The standard deviation of the carrier signal-to-noise density ratio, $\sigma(\mathrm{C/N_0})$, calculated within the elevation angle range $+1° < \epsilon < +2°$, is taken as proxy for the signal amplitude fluctuation.
>
> Each panel of fig. 2 includes information on the correlation; the Pearson and Spearman coefficients are quoted in the top and bottom line, respectively, (see, e.g., Press et al., 1992); the corresponding significance parameters are given in brackets. The numerical values indicate that $\langle dN/dz \rangle$ and the standard deviation of $\mathrm{C/N_0}$ are weakly to moderately correlated. With the exception of PRN 17 (top right panel) all calculated correlations are significant on the 5% level. The (negative) correlations range from $-0.17$ to $-0.40$. We note that ECMWF refractivity profiles below 1 km frequently exhibit strong gradients. Their inclusion into the calculation of $\langle dN/dz \rangle$ significantly decreases the correlations or even renders them insignificant.

The MPS simulations (see fig. 1) also suggests that the (negative) correlation between $\sigma\left(\mathrm{C/N_0}\right)$ and $\langle dN/dz \rangle$ weakens if elevation angles close to or below the horizon are included. The data displayed in fig. 3 confirms this prediction. It shows the correlation between $\sigma\left(\mathrm{C/N_0}\right)$ and $\langle dN/dz \rangle$, however in this case the elevation angle range used for the calculation of $\sigma\left(\mathrm{C/N_0}\right)$ is extended downwards to $-2°$. Comparison with fig. 2 shows that (with the exception of PRN 7 and the Pearson coefficient of PRN 18) the derived correlations are no longer significant, i.e. based on these results the null hypothesis cannot be rejected on the 5% significance level.

Finally, we note that the "GLESER" campaign raw data files have been supplied with the digital object identifier doi:10.5880/GFZ.2016.1.1.002 and are available through the DOI resolver `http://dx.doi.org/`. The data are supplemented with a set of documents describing the measurement data files and an archive containing the "OpenGPS" receiver software used during the measurement campaign.

[Figure]

Figure 2: Standard deviation of $C/N_0$ at elevation angles between $+1°$ and $+2°$ versus mean refractivity gradient for nine PRNs extracted from ECMWF (March–August 2014). For PRN 13, 14 and 23 one data point exceeds the axis limit of 4.2 dB Hz; its respective mean refractivity gradient is marked by an arrow. (Of course, these observations are included in the statistical analysis.) In the lower left corner of each panel correlation coefficients are given (top: Pearson's coefficient, bottom: Spearman's coefficient). The corresponding significance parameters are stated in brackets. Results from O/L channel B (green points) very closely agree with channel A data (red) and therefore almost completely mask the latter.

[Figure]

Figure 3: Same as fig. 2, however, the correlation analysis now includes all observations at elevations between $-2°$ and $+2°$. With the exception of PRN 7 (and the Pearson's coefficient for PRN 18) the derived correlations are no longer significant.

**Bibliography**

Anderson, K. D.: Determination of water level and tides using interferometric observations of GPS signals, J. Atmos. Ocean. Tech., 17, 1118–1127, doi:10.1175/1520-0426, 2000.

Anthes, R. A., Bernhardt, P. A., Chen, Y., Cucurull, L., Dymond, K. F., Ector, D., Healy, S. B., Ho, S.-P., Hunt, D. C., Kuo, Y.-H., Liu, H., Manning, K., McCormick, C., Meehan, T. K., Randel, W. J., Rocken, C., Schreiner, W. S., Sokolovskiy, S. V., Syndergaard, S., Thompson, D. C., Trenberth, K. E., Wee, T.-K., Yen, N. L., and Zeng, Z.: The COSMIC/FORMOSAT-3 mission: Early results, Bull. Am. Meteorol. Soc., 89, 313–333, doi:10.1175/BAMS-89-3-313, 2008.

Bevis, M., Businger, S., Chiswell, S., Herring, T. A., Anthes, R. A., Rocken, C., and Ware, R. H.: GPS meteorology: Mapping zenith wet delays onto precipitable water, J. Appl. Meteorol., 33, 379–386, doi:10.1175/1520-0450(1994)033<0379:GMMZWD>2.0.CO;2, 1994.

Braun, J., Rocken, C., and Ware, R.: Validation of line-of-sight water vapor measurements with GPS, Radio Sci., 36, 459–472, doi:10.1029/2000RS002353, 2001.

Businger, S., Chiswell, S. R., Bevis, M., Duan, J. P., Anthes, R. A., Rocken, C., Ware, R. H., and amd F. S. Solheim, M. E. T. V.: The promise of GPS in atmospheric monitoring, B. Am. Meteorol. Soc., 77, 5–18, doi:10.1175/1520-0477(1996)077<0005:TPOGIA>2.0.CO;2, 1996.

Cucurull, L., Derber, J. C., Treadon, R., and Purser, R. J.: Assimilation of Global Positioning System radio occultation observations into NCEP's global data assimilation system, Mon. Wea. Rev., 135, 3174–3193, doi:10.1175/MWR3461.1, 2007.

Dick, G., Gendt, G., and Reigber, C.: First experience with near real-time water vapor estimation in a German GPS network, J. Atmos. Sol.-Terr. Phy., 63, 1295–1304, doi:10.1016/S1364-6826(00)00248-0, 2001.

Elósegui, P., Davis, J. L., Jaldehag, R. T. K., Johansson, J. M., Niell, A. E., and Shapiro, I. I.: Geodesy using the Global Positioning System: The effects of signal scattering on estimates of site position, J. Geophys. Res.-Sol. Ea., 100, 9921–9934, doi:10.1029/95JB00868, 1995.

Emardson, T. R., Elgered, G., and Johansson, J. M.: Three months of continuous monitoring of atmospheric water vapor with a network of Global Positioning System receivers, J. Geophys. Res.-Atmos., 103, 1807–1820, doi:10.1029/97JD03015, 1998.

Foelsche, U., Scherllin-Pirscher, B., Ladstädter, F., Steiner, A. K., and Kirchengast, G.: Refractivity and temperature climate records from multiple radio occultation satellites consistent within 0.05%, Atmos. Meas. Tech., 4, 2007–2018, doi:10.5194/amt-4-2007-2011, 2011.

Gaylor, D. E., Lightsey, E. G., and Key, K. W.: Effects of multipath and signal blockage on GPS navigation in the vicinity of the International Space Station (ISS), Navigation, 52, 61–70, doi:10.1002/j.2161-4296.2005.tb01732.x, 2005.

Gleisner, H. and Healy, S. B.: A simplified approach for generating GNSS radio occultation refractivity climatologies, Atmos. Meas. Tech., 6, 121–129, doi:10.5194/amt-6-121-2013, 2013.

Gorbunov, M. E.: Radioholographic analysis of radio occultation data in multipath zones, Radio Sci., 37, 14–1–14–9, doi:10.1029/2000RS002577, 2002.

Grimault, C.: A multiple phase screen technique for electromagnetic wave propagation through random ionospheric irregularities, Radio Science, 33, 595–605, doi:10.1029/97RS03552, 1998.

Hagemann, S., Bengtsson, L., and Gendt, G.: On the determination of atmospheric water vapor from GPS measurements, J. Geophys. Res.-Atmos., 108, doi:10.1029/2002JD003235, 4678, 2003.

Healy, S. B. and Thepaut, J. N.: Assimilation experiments with CHAMP GPS radio occultation measurements, Q. J. R. Meteorol. Soc., 132, 605–623, doi:10.1256/qj.04.182, 2006.

Hofmann-Wellenhof, B.: Global Positioning System: Theory And Practice, Springer, Wien, New York, 2013.

Knepp, D. L.: Multiple phase-screen calculation of the temporal behaviour of stochastic waves, Proc. of the IEEE, 71, 722–737, 1983.

Kursinski, E. R., Hajj, G. A., Leroy, S. S., and Herman, B.: The GPS radio occultation technique, Terr. Atmos. Ocean. Sci., 11, 53–114, 2000.

Larson, K. M., Small, E. E., Gutmann, E., Bilich, A., Axelrad, P., and Braun, J.: Using GPS multipath to measure soil moisture fluctuations: initial results, GPS Solut., 12, 173–177, doi:10.1007/s10291-007-0076-6, 2008.

Liu, Y. and Xue, J.: Assimilation of GNSS radio occultation observations in GRAPES, Atmos. Meas. Tech., 7, 3935–3946, doi:10.5194/amt-7-3935-2014, 2014.

Martin, J. M. and Flatté, S. M.: Intensity images and statistics from numerical simulation of wave propagation in 3-D random media, Appl. Opt., 27, 2111–2126, doi:10.1364/AO.27.002111, 1988.

Melbourne, W. G.: Radio occultations using Earth satellites: A wave theory treatment, Wiley & Sons Ltd., 2004.

Parkinson, B. W. and Spilker, J. J., eds.: Global Positioning System: Theory & Applications, Volume I&II, Am. Inst. of Aeronautics and Astronautics, Washington, DC, 1996.

Poli, P., Healy, S. B., and Dee, D. P.: Assimilation of Global Positioning System radio occultation data in the ECMWF ERA-Interim reanalysis, Q. J. R. Meteorol. Soc., 136, 1972–1990, doi:10.1002/qj.722, 2010.

Press, W. H., Teucholsky, S. A., Vetterling, W. T., and Flannery, B. P.: Numerical recipies in C, The art of scientific computing, 2nd edition, Cambridge University Press, Cambridge, 1992.

Rennie, M. P.: The impact of GPS radio occultation assimilation at the Met Office, Q. J. R. Meteorol. Soc., 136, 116–131, doi:10.1002/qj.521, 2010.

Ringer, M. A. and Healy, S. B.: Monitoring twenty-first century climate using GPS radio occultation bending angles, Geophys. Res. Lett., 35, L05708, doi:10.1029/2007GL032462, 2008.

Rocken, C., Hove, T. V., and Ware, R.: Near real-time GPS sensing of atmospheric water vapor, Geophys. Res. Lett., 24, 3221–3224, doi:10.1029/97GL03312, 1997.

Schmidt, T., Wickert, J., and Haser, A.: Variability of the upper troposphere and lower stratosphere observed with GPS radio occultation bending angles and temperatures, Advances in Space Research, 46, 150–161, doi:10.1016/j.asr.2010.01.021, 2010.

Sokolovskiy, S. V.: Modeling and inverting radio occultation signals in the moist troposphere, Radio Sci., 36, 441–458, doi:10.1029/1999RS002273, 2001a.

Sokolovskiy, S. V.: Tracking tropospheric radio occultation signals from low Earth orbit, Radio Sci., 36, 483–498, doi:10.1029/1999RS002305, 2001b.

Steiner, A. K., Lackner, B. C., Ladstädter, F., Scherllin-Pirscher, B., Foelsche, U., and Kirchengast, G.: GPS radio occultation for climate monitoring and change detection, Radio Sci., 46, RS0D24, doi:10.1029/2010RS004614, 2011.

Tregoning, P., Boers, R., O'Brien, D., and Hendy, M.: Accuracy of absolute precipitable water vapor estimates from GPS observations, J. Geophys. Res.-Atmos., 103, 28,701–28,710, doi:10.1029/98JD02516, 1998.

Vedel, H. and Huang, X. Y.: Impact of ground based GPS data on numerical weather prediction, J. Meteorol. Soc. Jpn., 82, 459–472, doi:10.2151/jmsj.2004.459, 2004.

Ware, R., Alber, C., Rocken, C., and Solheim, F.: Sensing integrated water vapor along GPS ray paths, Geophys. Res. Lett., 24, 417–420, doi:10.1029/97GL00080, 1997.

Yunck, T. P., Liu, C.-H., and Ware, R.: A history of GPS sounding, Terr. Atmos. Ocean. Sci., 11, 1–20, 2000.

---

## Author Comment (AC2) · 9 Sep 2016

**Response to reviewer 2**

We thank the reviewer for his/her time, the thoughtful suggestions and helpful comments. Guided by the two reviewers' remarks the paper has been corrected and revised. In the following we provide point-by-point responses to the reviewer's suggestions. His/her remarks are set in italics, our answers are added in normal font. The revised text is indented.

*[...] However, a better experimental design and perhaps better site selection are needed to produce more impactful results. The main conclusions of this paper are not really new, at least qualitatively. In addition, one issue I have is with the implementation of its O/L tracking (see comment 3 below) and how this would affect the results quantitatively and how this would translate to the spaceborne case. Overall, I think the paper can be suitable for publication, provided that the comments are adequately addressed and that the paper is viewed as a "pathfinder" for future experiments.*

The main design criteria of the present study were twofold. First, the set-up should be based on a receiver software implementation available in source code format, to allow us complete control over all details of the open-loop signal tracking. Second, the project had to be accommodated within limited budget allocated to this activity. We agree that both, experimental design and measurement site selection could be improved and these modifications would most certainly lead to more relevant results. We intend to continue this line of research and hope to gather sufficient funding support for an optimized measurement set-up. The present submission is part of this effort.

And we fully agree that the "GLESER" campaign should be viewed as a pathfinder for future, more refined experiments with the objective to exploit the information content of low elevation GNSS data in the best possible way.

*(1) The paper should be made more succinct. In particular, the detailed description of the OpenGPS receiver does not offer much insight into understanding the data. The readers have to wade through too much background materials before getting to the results. I suggest significant shortening of Section 2.2 and Section 2.3. Another option is to move the materials to an Appendix.*

We agree that sections 2.2 and 2.3 are rather technical; in the revsied

manuscript both sections are shifted to the appendix.

*(2) L15:*

*it states that "vertical refractivity gradients ... correlate moderately well with observed signal amplitude fluctuations ..." However, Figure 16 shows that only 3 out of 9 cases "yield significant correlations". I suggest changing the wording of "correlate moderately well" to better represent the results.*

We accept the criticism, that our original approach, to understand the observed $C/N_0$ fluctuations, is too simplistic and is not convincing. Following a suggestion made by the first reviewer, Multiple Phase Screen (MPS) simulations of signal propagation at the low elevation angles have been performed to support the analysis of the $C/N_0$ variations.

The revised version of the paper contains the following new section as suggested by reviewer 1:

**Simulations**

In order to support the interpretation of the observed $C/N_0$ fluctuations we performed a series of Multiple Phase Screen simulations (Knepp, 1983; Martin and Flatté, 1988; Grimault, 1998). The propagation of a plane wave through the lower troposphere is modeled by a series of 500 non-equidistant phase screens ranging from the receiver location to a distance of 500 km. On each phase screen the wave suffers a phase delay determined by the interscreen distance $\Delta z$ and the refractivity height profile (Sokolovskiy, 2001)

$$N(h) = 400 \exp\left(\frac{-h}{h_s}\right) \left(1 - 0.05 \frac{2}{\pi} \mathrm{atan}\left(\frac{h - h_{tp}}{h_{zn}}\right)\right) \qquad (1)$$

with scale height $h_s = 8$ km, planetary boundary layer (PBL) top height $h_{tp}$ and PBL top transition zone $h_{zn} = 50$ m. The interaction of the wave with the ground surface is modelled by applying a raised-cosine filter with a 6 dB steepness of 25 m (i.e., within 25 m the filter weight decreases by 6 dB) at zero altitude. The phase screens extend from $-20$ km to $+20$ km with a 5 km wide raised-cosine filter applied at the upper and lower boundary to suppress spurious diffraction effects; the receiver altitude is taken to be 50 m. The variation of elevation angle between $-2°$ and $+2°$ is modelled by tilting the ground surface and its overlying atmosphere correspondly.

Results from four simulation runs are shown in fig. 1; it displays the normalized signal amplitude as a function of elevation angle. Signal absorption at the ground surface produces characteristic diffraction

[Figure]

Figure 1: Normalized signal amplitudes as a function of elevation angle derived from several Multiple Phase Screen simulations. Refractivities on the individual phase screens are calculated from an exponential profile furnished with a planetary boundary layer in the lower troposphere. Signal absorption at the surface is taken into account (red and blue lines), green and black lines show the result without ground absorption. Two boundary layers are modelled: a horizontal boundary layer top at 2 km altitude (red and black) and a layer top increasing from 1 km to 2 km between 30 and about 60 km distance from the receiver (blue and green). For legibility the red, green and blue lines are shifted by an additional +1 dB, +2 dB and +3 dB offset, respectively. For details see text.

patterns for elevation angels below 0° (red and blue lines). Without ground absorption the diffraction patterns almost disappear and the profiles resemble step functions expected from geometric optics (green and black). The simulations did not produce $C/N_0$ fluctuations for horizontally oriented PBL tops (parameterized by $h_{tp}$ in eqn. 1). However, if the top layer tilts towards the receiver, substantial signal deviations at elevation angles above 0° are observed. Fig. 1 illustrates this phenomenon for a PBL top layer ascending from 1 km at 30 km distance to 2 km at about 60 km (green and blue lines); below 30 km and above 60 km $h_{tp}$ remains fixed at 1 km and 2 km, respectively.

The MPS simulation results plotted in fig. 1 indicate ground effects below about 0° elevation angle (blue and red) and PBL-induced $C/N_0$ variations above about 0° (blue and green). The results suggest that these $C/N_0$ fluctuations are independent from each other and tend to be separated in elevation angle space. Finally, we note that the

addition of irregularities on spatial scales characteristic for turbulence to the refractivity profiles did not produce significant $C/N_0$ changes.

*(3) L245-255:*
*the O/L tracking models are extrapolated from the C/L tracking data at higher elevation angles. Thus one can argue that this is more similar to the "flywheel-ing" algorithm implemented on CHAMP and SAC-C and might not work well under some conditions (which the authors recognized, see L267-270). Please explain/justify why it is done this way. Is it possible to use an a priori O/L model as was done in COSMIC and Metop/GRAS?*

We agree that our open-loop implementation shares certain features of "fly-wheeling" signal tracking as decribed in Ao et al. (2003) On the other hand, questions regarding the precise differences between "fly-wheeling" and O/L are difficult to answer without access to implementation details, i.e. source code. We decided to classify our implementation as "open-loop" tracking, because the "OpenGPS" instrument outputs inphase/quadphase correlation sums together with NCO phases when O/L tracking is activated. CHAMP "fly-wheeling" raw data do not provide this information; thus, based on the CHAMP raw data analysis it cannot be unambiguously concluded that the carrier tracking loops were fully opened in "fly-wheeling" mode.

It is certainly possible for "OpenGPS" (and will be considered for future implementations) to use "fixed" a-priori O/L models as implemented COS-MIC and Metop/GRAS. However, it should be emphasized that owing to the closed architecture of the COSMIC and Metop/GRAS receivers (and hardware related differences) an exact emulation appears not feasible.

We suppose that a) details of the receiver firmware are relevant for the scientific data evaluation of RO observations and b) certain specific (admittedly detailed) questions on RO data can only be conclusively answered on the source code level of the tracking software. For this reason we make available the OpenGPS receiver source code (as part of the "GLESER" raw data archive). It can be accessed via the Digital Object Identifier (DOI) doi:10.5880/GFZ.2016.1.1.002.

*(4) Eqs (6)-(8):*
*is there a reference for calculating C/N0 this way? I am confused with Eq. (8) since this apparently yields a noise floor of 17 dB Hz that depends only on the integration time. Shouldn't this depend on the antenna gain, cable loss, etc.?*

Suitable references are Badke (2009); Kaplan (1996); Parkinson and Spilker (1996). Most likely the misunderstanding is caused by our usage of the term "noise floor" for the $C/N_0$ value of 17 dB Hz. $C/N_0$ is the carrier signal-tonoise density ratio and as such does not provide information on the noise measured by the receiver front-end. The value of 17 dB Hz represents the $C/N_0$ value with the signal completely blocked by the horizon. According to eqn. 6 this value depends only on the chosen coherent integration time $T_c$. In the revised version the misleading term "noise floor" has been removed.

*(5) Table 1:*
*PRN28 yields anomalously low (about 35%) O/L enhancement. Any idea why?*

Our open-loop implementation rests on the assumption that for elevation angles between 0 and $+2°$ signal tracking is stable without strong $C/N_0$ fluctuations. The low O/L enhancement for PRN28 suggests that this assumption is invalid in this instance.

The revised version of paper includes the paragraph (figure number 4, mentioned in the text, refers to the submitted paper, not the present document):

> The low O/L enhancement values for PRN 28 might be caused by signal reflections at the water surface of Templiner lake (see fig. 4 at about 260° azimuth angle). As described in the appendix (subsection A2), the O/L model is initialized within the elevation angle range between 0° and $+2°$. It appears feasible that surface reflections induce $C/N_0$ fluctuations at these elevation angles degrading the quality of the O/L model and thereby causing poor O/L performance.

*(6) Fig. 15:*
*I suggest more distinct colors for the light blue and dark blue lines. I have a hard time distinguishing them. It is also hard to tell the actual values of frequency offsets from these plots. I think it would be useful to have a summary table of mean and standard deviation of $\Delta f_{obs}$ as a function of C/N0 that average over all 9 PRNs.*

In the revised paper a better line coloring is used (see fig. 2) and a summary table showing the mean and standard deviation of $\Delta f_{\mathrm{obs}}$ as a function of $C/N_0$ is included (see table 1; the figure numbers mentioned in the figure and table captions refer to the submitted paper, not the present document).

*(7) L522:*
*Is the refractivity based on ECMWF at a grid point closest to the receiver? How about temporal differences? How does the refractivity vary along the ray paths? Besides vertical refractivity, horizontal inhomogeneity and small-scale irregularities (turbulence) can also lead to strong signal fluctuations. These effects could perhaps explain the lack of correlations for many of the PRNs.*

Table 1: Mean and standard deviations of $\Delta f_{\mathrm{obs}}$ (center column) for nine values of the carrier signal-to-noise density ratio $C/N_0$ between 10 and 50 dB Hz; averaging bin size is 5 dB Hz. The statistics is based on all PRNs shown in fig. 15. The third column gives the corresponding result neglecting frequency deviations larger than 40 Hz.

| $C/N_0$ [dB Hz] | $\Delta f_{\mathrm{obs}}$ [Hz] | $\Delta f_{\mathrm{obs}}^{<40\,\mathrm{Hz}}$ [Hz] |
|---|---|---|
| 10 | $8.23 \pm 18.08$ | $4.51 \pm 14.01$ |
| 15 | $5.48 \pm 15.47$ | $1.85 \pm 9.68$ |
| 20 | $2.75 \pm 11.50$ | $0.50 \pm 5.68$ |
| 25 | $1.28 \pm 7.89$ | $0.19 \pm 3.29$ |
| 30 | $1.23 \pm 7.71$ | $0.08 \pm 1.96$ |
| 35 | $1.39 \pm 8.17$ | $0.06 \pm 1.38$ |
| 40 | $0.95 \pm 6.78$ | $0.04 \pm 1.12$ |
| 45 | $0.62 \pm 5.47$ | $0.05 \pm 1.28$ |
| 50 | $0.56 \pm 5.23$ | $0.06 \pm 1.68$ |

In the orginal version of the paper the closest ECMWF grid point was selected, which is about 42 km south of the receiver location. In the revised analysis an alternative approach was followed and grid points oriented towards western directions were chosen (see below).

Following a suggestion by reviewer 1 we performed provisional Multiple Phase Screen (MPS) simulations to support the interpretation of the observed $C/N_0$ profiles, in particular the observed $C/N_0$ fluctuations. The revised paper includes an additional section and figures describing the MPS results (see response to specific comment below). We note, that our MPS simulations failed to produce significant $C/N_0$ fluctuations if irregularities on spatial scales characteristic for turbulence were added to the refractivity profiles.

The revised section describing the correlation between the observed $C/N_0$ fluctuations and ECMWF-derived refractivity gradients reads as follows:

The MPS simulations (fig. 1) suggest that at negative elevation angles diffraction effects caused by the ground surface dominate the observed $C/N_0$ fluctuations. At higher elevations atmospheric multipath seems to be more relevant. This hypothesis is tested for the six month time period from March to August 2014 by correlating the standard deviation of $C/N_0$ between elevation angles of $+1°$ and $+2°$ with the mean refractivity gradient $\langle dN/dz \rangle$. The calculation of $\langle dN/dz \rangle$ is restricted to the altitude range from 1 to 3 km. Fig. 3 shows the results for nine

PRNs.

The vertical refractivity profiles $N(z)$ are extracted from European Centre for Medium-Range Weather Forecasts (ECMWF) meteorological fields. Their horizontal resolution is $1° \times 1°$ (about 110 km in meridional and 69 km in zonal direction at the receiver location) with 137 height levels ranging from 0 to about 80 km; the averaging interval of 1 to 3 km corresponds to about 13 vertical height levels. For signal azimuth angles less than $-90°$ (west to south-west) the refractivity profile is extracted from ECMWF grid point (52°N, 12°E), about 84.4 km south-west of the observation site ($-119.8°$ true bearing). For azimuth angles greater than $-90°$ (west to north-west) the ECMWF grid point (53°N, 12°E) is selected, which is located about 99.8 km in the north-western direction ($-45.8°$ true bearing). The standard deviation of the carrier signal-to-noise density ratio, $\sigma(C/N_0)$, calculated within the elevation angle range $+1° < \epsilon < +2°$, is taken as proxy for the signal amplitude fluctuation.

Each panel of fig. 3 includes information on the correlation; the Pearson and Spearman coefficients are quoted in the top and bottom line, respectively, (see, e.g., Press et al., 1992); the corresponding significance parameters are given in brackets. The numerical values indicate that $\langle dN/dz \rangle$ and the standard deviation of $C/N_0$ are weakly to moderately correlated. With the exception of PRN 17 (top right panel) all calculated correlations are significant on the 5% level. The (negative) correlations range from $-0.17$ to $-0.40$. We note that ECMWF refractivity profiles below 1 km frequently exhibit strong gradients. Their inclusion into the calculation of $\langle dN/dz \rangle$ significantly decreases the correlations or even renders them insignificant.

The MPS simulations (see fig. 1) also suggests that the (negative) correlation between $\sigma(C/N_0)$ and $\langle dN/dz \rangle$ weakens if elevation angles close to or below the horizon are included. The data displayed in fig. 4 confirms this prediction. It shows the correlation between $\sigma(C/N_0)$ and $\langle dN/dz \rangle$, however in this case the elevation angle range used for the calculation of $\sigma(C/N_0)$ is extended downwards to $-2°$. Comparison with fig. 3 shows that (with the exception of PRN 7 and the Pearson coefficient of PRN 18) the derived correlations are no longer significant, i.e. based on these results the null hypothesis cannot be rejected on the 5% significance level.

*(8)*
*Do you expect local environmental effects (e.g., local multipath) and ionospheric conditions to have significant impact on the measurements?*

Ionospheric refraction and multipath certainly has an effect on the observed code delays and carrier phases. It is very well possible, that these phenomena cause $C/N_0$ fluctuations as well. Since the "OpenGPS" receiver is a single frequency device, there is unfortunately no way extract a definite answer from the measurement data. The neglect of both aspects, local multipath and ionospheric dispersion may partly explain non-perfect correlation listed in fig. 3.

In the revised paper the following sentence is added to the conclusions:

> The present study did not address potential contributions of the ionospheric signal propagation and/or local multipath to the observed $C/N_0$ fluctuations. These are important issues that need to be addressed in future work preferably using dual frequency receivers.

Finally, we note that the "GLESER" campaign raw data files have been supplied with the digital object identifier doi:10.5880/GFZ.2016.1.1.002 and are available through the DOI resolver `http://dx.doi.org/`. The data are supplemented with a set of documents describing the measurement data files and an archive containing the "OpenGPS" receiver software used during the measurement campaign.

[Figure]

Figure 2: Same as fig. 10, however, showing the difference between the two observed frequencies obtained from O/L channel A and B as a function of the mean signal-to-noise density ratio. Mean and 1-$\sigma$ standard deviations, calculated from $C/N_0$ bins 2.5 dB Hz wide, are marked in green. The fraction of data points exceeding $\Delta f_{\mathrm{obs}} > +40$ Hz is indicated as $\rho_{40\,\mathrm{Hz}}$. The result of the statistical analysis excluding this subset still exhibits a positive bias if $C/N_0 \overset{<}{\sim} 30$ dB Hz (red).

[Figure]

Figure 3: Standard deviation of $C/N_0$ at elevation angles between $+1°$ and $+2°$ versus mean refractivity gradient for nine PRNs extracted from ECMWF (March–August 2014). For PRN 13, 14 and 23 one data point exceeds the axis limit of 4.2 dB Hz; its respective mean refractivity gradient is marked by an arrow. (Of course, these observations are included in the statistical analysis.) In the lower left corner of each panel correlation coefficients are given (top: Pearson's coefficient, bottom: Spearman's coefficient). The corresponding significance parameters are stated in brackets. Results from O/L channel B (green points) very closely agree with channel A data (red) and therefore almost completely mask the latter.

[Figure]

Figure 4: Same as fig. 3, however, the correlation analysis now includes all observations at elevations between $-2°$ and $+2°$. With the exception of PRN 7 (and the Pearson's coefficient for PRN 18) the derived correlations are no longer significant.

**Bibliography**

Ao, C. O., Meehan, T. K., Hajj, G. A., Mannucci, A. J., and Beyerle, G.: Lower-Troposphere Refractivity Bias in GPS Occultation Retrievals, J. Geophys. Res., 108, 4577, doi:10.1029/2002JD003216, 2003.

Badke, B.: What is $C/N_0$ and how is it calculated in a GNSS receiver?, InsideGNSS, 4, 20–23, 2009.

Grimault, C.: A multiple phase screen technique for electromagnetic wave propagation through random ionospheric irregularities, Radio Science, 33, 595–605, doi:10.1029/97RS03552, 1998.

Kaplan, E. D.: Understanding GPS: Principles and applications, Artech House, Boston, London, 1996.

Knepp, D. L.: Multiple phase-screen calculation of the temporal behaviour of stochastic waves, Proc. of the IEEE, 71, 722–737, 1983.

Martin, J. M. and Flatté, S. M.: Intensity images and statistics from numerical simulation of wave propagation in 3-D random media, Appl. Opt., 27, 2111–2126, doi:10.1364/AO.27.002111, 1988.

Parkinson, B. W. and Spilker, J. J., eds.: Global Positioning System: Theory & Applications, Volume I&II, Am. Inst. of Aeronautics and Astronautics, Washington, DC, 1996.

Press, W. H., Teucholsky, S. A., Vetterling, W. T., and Flannery, B. P.: Numerical recipies in C, The art of scientific computing, 2nd edition, Cambridge University Press, Cambridge, 1992.

Sokolovskiy, S. V.: Modeling and inverting radio occultation signals in the moist troposphere, Radio Sci., 36, 441–458, doi:10.1029/1999RS002273, 2001.